# Benefit sharing in international rivers: A Q-methodology study of regional understanding and perception in Asia

Lei Xie[1], Lu Xu[2]*, Qi Yu[3]

**1** International Hydropower Association, London, United Kingdom, **2** Law School, Lancaster University, Lancaster, United Kingdom, **3** PhD candidate, Shandong University, Jinan, China

* lu.xu@lancaster.ac.uk

**Data Availability Statement:** All relevant data are within the paper and its Supporting Information file.

**Funding:** Lei Xie received funding from Shandong University (www.sdu.edu.cn, Funding Reference

## Abstract

This study aims to identify and examine the different perception of benefit sharing in the sharing of international rivers in China, South Asia and Southeast Asia. Using the Q-Method, this study undertakes an in-depth analysis of the views of 35 experts of the field on hydrodiplomacy, international water law, benefit sharing and ecological benefits compensation. The results of the quantitative and qualitative analysis help to innovatively identify three streams of views among the participants, respectively described as supporters, idealists and pragmatists, each displaying strong geographical association to the three Asian regions. Supporters from Southeast Asia and pragmatists from China share much common ground on issues such as the types of benefits to share in international rivers, potentially providing the conceptual foundation for international cooperation. Idealists from South Asia prioritize and emphasise the role and importance of environmental benefits and ecological protection, yet differ greatly from the others on practical issues such as the inclusion of direct economic benefits and ecological compensation for using resources. This study contributes to the understanding of the theory and practice of benefit sharing in international rivers, as well as providing new perspectives to the interpretation and practice of hydrodiplomacy in Asian regions.

## 1. Introduction

Benefit sharing is a buzz word of contemporary discourse and discussion over the sharing of international rivers as countries seek to manage such shared resources. The idea of benefit sharing is rooted in viewing water as a type of resource that possesses multiple potential values to benefit from. In principle, benefit-sharing arrangements utilize water resources in ways that take into consideration economic, policy and environmental concerns and facilitate the efficient use of water resources by all riparian states of transboundary rivers [1, 2]. Nevertheless, both conceptually and practically, there is ambiguity over its role and recognition by current international law [3]. Benefit sharing is not explicitly recognized under any formal legal instrument [4].

No.61060082035302). The funder had no role in study design, data collection and analysis, decision to publish, or preparation of the manuscript.

**Competing interests:** The authors have declared that no competing interests exist.

Despite such uncertainty and ambiguity, there are notable examples of benefit sharing of international rivers globally, such as in water basins in the Andes in South America [1], the Columbia River in North America [5, 6], the Senegal basin in Africa [7] and more recently the Mekong River in Asia [8]. At the same time, there are considerable challenges and difficulties, especially political risks, in implementing this approach to other regions and transboundary waters [7]. The difficulty for cooperation is particularly notable in the developing world, such as among China, India and their neighbouring states over international rivers [9, 10]. Although there have been several studies that explain the nature of conflicts as well as obstacles in the cooperation between China and India from different perspectives [11, 12], few have approached the topic from aspects of conceptual difficulties and differences of benefit sharing, which may well play an important role in this context.

In order to understand the different perspectives of the understanding of benefit sharing in parts of Asia, this present study undertakes a Q methodology survey of experts from China, South Asia and Southeast Asia. It seeks to answer the following research questions. What benefits are to be shared and how important is benefit sharing perceived by individual experts as an established principle in international water governance? Is the understanding of benefit sharing affected by geographical association? In what ways does benefit sharing provide incentives for countries to develop diplomatic mechanisms and reconcile differences in their potentially competing interests over shared water resources? The empirical findings help to identify different streams of views on benefit sharing and related concepts in the region, facilitating explanation and interpretation of the varying degrees of cooperation among the riparian neighbours.

This paper is organized as follows. Section 2 undertakes a literature review of the current theories and conception of benefit sharing and related issues. Section 3 explains the Q methodology and its practical implementation in the present study. Section 4 presents the empirical finding from the Q methodology survey, including the analysis of three different streams of views on benefit sharing among the expert participants. Section 5 evaluates the implications of these findings and the different perspectives for the practical implementation of benefit sharing over international rivers in the regions concerned. Section 6 concludes with a summary of the significance and value of this study.

## 2. Conceptualization of benefit sharing

### 2.1 Contextualizing benefit sharing in Asia

Conceptually, conserving rivers and protecting riparian community interests are established environmental activities that have been widely recognized around the world in policy practices over water management. Even before the theorization of benefit sharing, the international community has focused on developing and refining principles of shared management in the world's international waterways in the 20th century [5]. Since Sadoff and Grey [13] first proposed an analytical framework describing the various benefits from cooperation on international rivers, optimistic views advocate that benefit sharing serves the protection of ecosystem at its core, which is beneficial to livelihood of communities across shared river. The actual label of benefit-sharing in this context could be traced to the term used in the realm genetic resources and biodiversity [7]. The concept of benefit sharing has been defined as any action designed to change the allocation of costs and benefits associated with cooperation, which requires some sort of redistribution or compensation in most cases [14].

However, competing and potentially conflicting interests may be derived from water resources, which often determine the nature of interaction and extent to which benefits derived from rivers could extend to the riparian users [15]. Environmentally conscious

scholars have strongly argued for prioritizing the ecosystem over state interests in managing transboundary river basins [16]. For countries in the Global South, in particular, exploring economic interests from water is often preferred by state actors, while considerations of the ecosystem sometimes become of secondary importance [17–19]. For some, water exists as an integrated part of the ecosystem, which represents an extended system that includes other natural systems such as wildlife habitat, and biodiversity [20, 21]. In contrast, others are concerned over livelihood security as millions of residents rely on fisheries for their livelihood [22], especially the livelihood of the poor [23]. Hence priority in water management emphasizes the protection of biodiversity, with an aim to protect productivity, such as ensuring fish habitats and their migration and breeding cycles not to be irreparably altered and possibly destroyed [24, 25]. Incidentally, government's focuses on exploiting the economic values of water resources are often identified in China [11] and India respectively [26–28]. Conflicting views exist domestically and internationally on how water resources should be used [29, 30], which in turn afford much importance to hydrodiplomacy.

Hydrodiplomacy refers to 'a process that promotes and relies on mutual cooperation and the idea of shared-benefits', and it 'can be seen as an operative end-product of the process of water diplomacy, with the very aim of the cooperation defined by the diplomatic process' [31]. Benefit sharing is viewed as the cornerstone in promoting hydrodiplomacy [31]. Commentators indicate that various benefits are to derive from different sectors in relation to water, the idea of sharing then allows countries to develop water sharing regimes [32, 33]. In Asian regions where riparian countries display dependence of water resources for energy, food and livelihood, which extend beyond water ecology, collectively exploring water's inter-sectoral interests such as hydropower presents an appealing option [30, 34, 35]. Nevertheless, the practical approach of developing hydrodiplomacy, based on the consent of sharing benefits from various water related sectors, remains undefined. In regions that experience power relations conflicts, it is often observed that power remains an identified obstacle that impact countries' efforts of seeking fairness or justice in realizing their right over shared water, especially when they experience different social, economic and political systems [9].

Furthermore, it is observed that international law currently offers very little help to countries' effort in the development of hydrodiplomacy or framework of benefit sharing, especially in this part of Asia. Of all the countries of East, Southeast and South Asia in this discussion, only Vietnam is a signatory to the 1997 Convention on the Law of the Non-Navigational Uses of International Watercourses of the United Nations. Although the 1992 Convention on the Protection and Use of Transboundary Watercourses and International Lakes, which started as a regional treaty under the auspice of the United Nations Economic Commission for Europe, is now a global instrument that attracted the accession of several Central Asia countries, it has made little impact in this region. Meanwhile, not only do criticism of the 1997 Watercourses Convention of it being full of vague, broad and general terms that made it difficult for countries to arrive at mutually acceptable understanding still hold much influence [36], developments such as the concept of benefit sharing in the past two decades also mean that the international legal instruments often represent an overly narrow approach to the construction of rights, obligations and benefits of riparian states, which further limits their practical effectiveness.

Existing scholarship emphasizes that different stakeholders play significant roles in determining the distribution of benefits [17, 18, 33]. Analysing the management of transboundary river basins in the developing world, Mirumachi [18] opines how individual actors influence policy outcomes and diplomatic cooperation in transboundary water sharing, such as politicians, civil society, academics, the media and so on. However, gap exists as to how individuals perceive water benefits and how to share them among nation states. Both normative and

objective mind frames exist when viewing how shared water resources should be managed and how inter-governmental cooperation can be conducted [37].

In conceptualizing benefit sharing principle, two key elements can be found that specify how the principle is defined and implemented. These are the types of benefits to be shared and ecological compensation as a mechanism that facilitates the practice of sharing these benefits.

## 2.2 Definitions of 'benefits' in shared rivers

In the context of sharing international rivers, the four types of benefits first identified by Sadoff and Grey [13] are widely used as the basis for discussion. These include environmental benefits to the river, direct economic benefits from the river, political benefits from reducing costs because of the river, and indirect benefits beyond the river. More specifically, environmental benefits promote the sustainable development of river basins; direct economic benefits from the river emphasizes water as an output, but not an input, by providing such commodities as electricity, food, and environmental services [7]. The maximization of economic benefits is legitimate only when environmental damage and disruption are avoided. Political benefits represent both symbolic and actual advantages for countries when they reach agreements on how best to benefit from collectively using the shared water resources. Indirect benefits beyond the river include a range of benefits often surpassing environmental and economic gains, including for instance, trade and regional development [7, 13, 38, 39].

Among these different types, environmental benefits underpin all other benefits that can be derived [9, 13]. Yet developing countries have the tendency to prioritize economic gains while sacrificing ecological benefits [8, 40]. Questions remain as to whether all these benefit types are accepted in practice, whether environmental benefits are perceived to form the basis of other benefits as envisaged by theorists, or whether nuance is needed in finding common ground for countries to establish benefit sharing mechanisms. Any commonly agreed benefit sharing principle will be particularly valuable to the development of hydrodiplomacy in the Global South [4, 33], where conflicting interests exist over the use of shared water resources and where countries have already shown disagreements over whether to adopt international water norms [11].

## 2.3 Ecological compensation

To facilitate the implementation of benefit sharing principle, ecological compensation is introduced as a mechanism encapsulating financial compensation, material compensation and policy support, that offers conditional payments to resource providers for their environmental gain services and focuses on achieving the goal of ecological environmental protection by incentives [41]. It should be noted that scholarship has placed strong emphasis on the application of benefit sharing principle, recognizing a potential trade-off between different services and profits desired from shared water resources [42]. Under the ecological benefit compensation principles, beneficiary countries that have paid compensation will further appreciate the preciousness of such resources and the ecological environment that they had to pay to preserve [43]. At the same time, ecological compensation principle also serves to resolve potential generational injustice that may occur in individual states when they contribute to the protection of ecological resources for the collective benefits across the region [43].

## 3. Methodology

In order to analyse the different perspectives in the understanding of the concept of benefit sharing, this study employs the Q methodology. The Q methodology enables the empirical study of human subjectivity, referring to the communication of a personal point of view, and

brings qualitative research into the quantitative realm [44]. The Q methodology has been widely used in research in psychology, sociology, political science and other fields [45]. There are multiple studies based on the Q methodology examining environmental governance and disputes [12, 46, 47], demonstrating its capability in facilitating a more practical understanding of the subjective perspectives in these areas of considerable complexity and conflicting information.

The Q methodology is particularly suited to this study because it requires a much smaller number of participants than other quantitative analysis, making it possible for the present study to solely focus on the opinions of experts in the field within the geographic locations of China, South and Southeast Asia. It enables rigorous and objective analysis and comparison of the subjective understanding of these experts and reveals often contradictory views on important issues that are often overlooked in the construction of doctrinal and conception frameworks. Meanwhile, the limitations of such Q methodology study also dictate that this study makes to attempt to assess the merits of the respective views beyond the fact that they reflect different aspects of the collective mentality among the group or subgroups of participants. Nevertheless, it is believed that the value of identifying previously unknown and often controversial viewpoints among regional experts, in the effort to build a more comprehensive understanding and a wider basis for consensus, would far outweigh such constraints of methodology.

This study follows the standard steps of Q methodology [45], summarized as follows:

1. Creation of a Q concourse, which is a set of statements that are broadly representative of the discourse on the topic of interest to the researchers.
   Based on the examination of existing literature on the relevant topics and interview responses from a number of experts, the present study established a set of 37 statements broadly representative of the discourse on a number of issues ranging from hydrodiplomacy, international water law, benefit sharing to ecological benefits compensation.

2. Administration of the Q-sort to persons whose perspectives on the topic are of interest to the researchers. Each person sorts the statements in a quasi-normal distribution, placing each statement in relation to all other statements on the basis of the instructions given (see Fig 1). The selection of persons to complete the Q-sort is purposeful, designed to include people whose opinions are of practical or theoretical interest.
   The study was approved by Shandong University School of Political Science and Public Administration as complying with all rules and criteria of the University, including for research involving human participants. All participants agreed to take part online or over the phone, and gave informed consent to use their answers and Q-sort results in the research and to report and publish any finding on the basis of anonymity. Each participant was connected online to one member of the research team during administration of the Q-sort. The researcher immediately asked participants for feedback after completion of the sort, including whether the concourse reflected an unbiased overall view, i.e. whether the statements left at "0" would match the neutral viewpoint of the participant.
   The present study obtained the Q-sort response of 35 experts in relevant fields from Bangladesh, Cambodia, China, India, Laos, Myanmar and Thailand. The number of participants is kept below the number of statements in the Q-sort, in accordance with general Q methodology theory [48]. Expert participants in this study include academics, experts from NGOs and think tanks, government officials in the water sector, and committee members of relevant international bodies.

3. Correlation of the completed sorts and the factor analysis of the correlation matrix for the purpose of identifying clusters or groups of participants who sorted their statements similarly.

| Most Disagreed | | Neutral | | | Most Agreed | |
|---|---|---|---|---|---|---|
| -3 | -2 | -1 | 0 | 1 | 2 | 3 |

**Fig 1. Q-Sort Grid used.**

Using PQ Method software, this study first performed a centroid factor analysis. It then conducted the principal components factor analysis, using varimax rotation on seven factors. Following Kaiser's 'eigenvalue greater than 1' method [49], 13 factors had eigenvalue greater than 1. However, the first seven factors collectively explained more than 60% of the study variance, while the other six each contribute to less than 5%. Therefore, only the first seven were rotated in this study.

The main results are presented in Table 1. In selecting factors for final interpretation [48], out of the various loadings in the results, three factors each with five or more loadings represented clear clustering of perspectives among the participants, which is of greater interest in Q-method study. After excluding confounded (significant loading on more than one studied factor) and non-significantly loaded Q-sorts, these three factors cumulatively explain 57% of the study variance. Other four factors with only two or three loadings are excluded from further analysis due to the lack of such concentration and scale. It is worth noting that several studies chose to focus on three or fewer factors in their analysis [50–52], including in the area of hydrodiplomacy [12].

4. Close examination of the weighted average sorts of the different groups of participants (the factors in the factor analysis) to identify the attitudes that characterize each of them and which cause them to differ from each other. This is analysed in Section 4 below.

## 4. Findings: Interpretation of factors and geographic association

The Q methodology study aims to identify and explain the perspective, viewpoint or attitude that is represented by each factor [48]. The interpretation of such perspectives draws on the quantitative data from the factor analysis of Q-sorts above, qualitative information gathered from interviews with participants, as well as the literature that formed the basis of the Q-statements. The process focuses on statements that have extreme factor scores and statements that help to distinguish between multiple factors, as these statements tend to reflect on the core values of each perspective. In the following analysis, the number in the parentheses refer to the 37 statements (S1-S37, see Table 2). The streams of views behind the three key factors (Factors 2, 5 and 6 in Table 1) are respectively labelled as supporters,

**Table 1. Notable factors and geographical association (X denotes significant factor loadings).**

| QSORT | | Factor1 | Factor 2 | Factor 3 | Factor 4 | Factor 5 | Factor 6 | Factor 7 |
|---|---|---|---|---|---|---|---|---|
| China | 1 | 0.0807 | **0.7677X** | 0.1613 | 0.1511 | 0.0174 | **0.3842X** | -0.1231 |
| | 2 | 0.3235 | 0.2545 | 0.3189 | 0.2284 | 0.3262 | **0.0449X** | -0.2700 |
| | 3 | -0.1687 | 0.2187 | 0.4939 | 0.4556 | 0.1573 | **0.0042X** | -0.2982 |
| | 4 | -0.1986 | 0.0485 | **0.6458X** | -0.1984 | -0.1118 | **-0.1731X** | 0.0008 |
| | 5 | 0.4507 | -0.2324 | 0.3345 | -0.3394 | -0.0122 | **0.1242X** | 0.0631 |
| | 6 | 0.0853 | -0.1333 | -0.0240 | -0.3124 | **0.5811X** | **0.2482X** | 0.0453 |
| | 7 | 0.1316 | 0.0740 | 0.1773 | **-0.6128X** | 0.2175 | -0.0181 | -0.1340 |
| | 8 | 0.1095 | 0.1162 | **0.7581X** | 0.0572 | 0.1857 | -0.0447 | 0.0853 |
| | 9 | 0.4455 | -0.0228 | 0.3223 | -0.0361 | -0.4645 | -0.0685 | 0.2307 |
| | 10 | 0.3766 | -0.2432 | 0.3087 | 0.3993 | 0.1113 | 0.1256 | 0.1006 |
| | 11 | 0.2568 | 0.3829 | 0.0112 | 0.0355 | 0.0659 | -0.2679 | 0.0281 |
| | 12 | 0.5381 | 0.0413 | 0.1268 | -0.1921 | -0.1674 | -0.1347 | 0.2931 |
| | 13 | 0.2151 | -0.0392 | 0.0018 | -0.0009 | -0.2275 | 0.5136 | 0.0452 |
| | 14 | 0.1995 | 0.4398 | -0.0904 | 0.3945 | -0.0851 | -0.5159 | 0.1770 |
| Southeast Asia | 15 | 0.1334 | 0.0298 | 0.2253 | **0.6242X** | 0.2181 | 0.1696 | 0.1393 |
| | 16 | 0.3603 | **0.7406X** | 0.0187 | 0.0401 | 0.0061 | -0.2076 | 0.1408 |
| | 17 | -0.0106 | 0.1204 | 0.1445 | -0.1215 | 0.0831 | -0.0400 | **0.7978X** |
| | 18 | 0.0193 | **0.5600X** | 0.0086 | 0.3826 | -0.1391 | -0.0031 | 0.1368 |
| | 19 | **0.9048X** | 0.0608 | -0.0993 | 0.0291 | 0.1088 | 0.1542 | 0.0494 |
| | 20 | 0.3158 | -0.1005 | -0.1260 | 0.0483 | 0.1803 | -0.5469 | 0.2827 |
| | 21 | 0.1432 | **-0.5487X** | -0.2329 | 0.2014 | -0.3439 | 0.0423 | 0.1435 |
| | 22 | -0.0249 | 0.1019 | -0.2174 | -0.1147 | 0.0229 | 0.1564 | 0.0785 |
| | 23 | 0.0653 | 0.0740 | -0.0503 | -0.0324 | **-0.7228X** | 0.2372 | 0.0709 |
| | 24 | 0.0825 | -0.0876 | 0.1259 | **-0.6402X** | 0.1427 | 0.0987 | 0.3851 |
| | 25 | 0.0776 | **0.4824X** | -0.4021 | -0.1123 | 0.1828 | 0.0334 | 0.0483 |
| | 26 | 0.0924 | 0.2502 | 0.1908 | 0.2022 | 0.1129 | 0.1270 | 0.2183 |
| India | 27 | 0.2146 | -0.1676 | 0.2720 | 0.0870 | **0.6142X** | -0.2736 | 0.0674 |
| | 28 | -0.1185 | **0.6488X** | 0.0492 | -0.0801 | -0.0594 | 0.0237 | -0.1702 |
| | 29 | 0.3268 | 0.2147 | -0.0139 | 0.0188 | **0.6047X** | -0.0326 | 0.2233 |
| | 30 | 0.0361 | -0.2381 | -0.0843 | 0.3523 | 0.0676 | 0.0971 | **0.7350X** |
| | 31 | 0.0532 | 0.0319 | **0.5630X** | -0.0535 | 0.3267 | 0.1262 | 0.1210 |
| | 32 | 0.1203 | 0.1071 | -0.1455 | 0.1425 | 0.0766 | 0.7088 | 0.1198 |
| | 33 | **0.9048X** | 0.0608 | -0.0993 | 0.0291 | 0.1088 | 0.1542 | 0.0494 |
| | 34 | **0.6683X** | 0.1816 | 0.0454 | -0.0779 | 0.0944 | -0.0970 | -0.4131 |
| | 35 | 0.0079 | 0.2264 | 0.2178 | -0.1147 | **0.5180X** | 0.0977 | 0.1554 |
| Total | | *3* | *6* | *3* | *3* | *5* | *6* | *2* |

idealists and pragmatists (see Table 3), with their most pronounced views of individual statements further displayed in Table 4.

## 4.1 Supporters

The first stream of views firmly supports the key principles of benefit sharing, as well as international law and hydrodiplomacy more generally. These participants see as imperative the need to consider all water stakeholders as a whole, in order to ensure the maximization of benefits in transboundary rivers and that all stakeholders will get the best benefit (S14). They strongly reject the notion that benefit sharing has not become an explicit obligation of any

**Table 2. Opinions of Q statements by perspective grouping (shaded cells highlight the five most strongly supported views of each groups).**

| NO. Q | Q Statements | F2 (Supporters) | | F5 (Idealists) | | F6 (Pragmatists) | |
|---|---|---|---|---|---|---|---|
| | | Score | Order | Score | Order | Score | Order |
| 1 | Due to the regional particularity of transboundary rivers, there is often considerable disparity between the rights and obligations for each individual riparian state. | -1.28 | 34 | -1.12 | 31 | -0.32 | 26 |
| 2 | The allocation of water right is closely related to a country's national strategy. | 0.31 | 15 | -1.38 | 34 | -0.15 | 23 |
| 3 | The right to access and use of international rivers should not be referred to as water right. Instead, it should be regarded as part of the sovereignty. | -1.88 | 37 | -2.30 | 37 | -1.80 | 35 |
| 4 | International water rights over shared transboundary rivers are specific sharing schemes negotiated among riparian states, rather than a certain kind of right or interest that is well-documented, regulated, and predictable. | 0.07 | 19 | -1.31 | 33 | -2.26 | 36 |
| 5 | Among the conflicts that arose from countries that share international rivers, the more riparian states there are, the greater the differences of interests. | 0.12 | 18 | -1.40 | 35 | -0.31 | 25 |
| 6 | Among the conflicts that arose from countries that share international rivers, the greater the differences of national conditions (e.g. socio-economics) there are, the more claims for interests there will be. | -0.24 | 25 | -0.22 | 24 | 0.21 | 19 |
| 7 | Among the conflicts that arose from countries that share international rivers, the more cooperation mechanisms there are, the harder it is to coordinate interests. | -1.69 | 35 | -0.98 | 30 | -2.48 | 37 |
| 8 | Generally speaking, the principle of equitable and reasonable utilization in the Convention on the Law of the Non-Navigational Uses of International Watercourses benefits upstream countries, while the principle of not causing significant harm benefits downstream countries. | -1.10 | 32 | -0.70 | 28 | 0.27 | 17 |
| 9 | The Convention on the Law of the Non-Navigational Uses of International Watercourses fails to strictly uphold the principle of fairness, in favouring the interests in water resources of the downstream countries and imposing excessive obligations on the upstream countries. | -0.23 | 24 | -1.17 | 32 | -0.24 | 24 |
| 10 | In several aspects, such as the obligations for notification or consultation, the Convention on the Law of the Non-Navigational Uses of International Watercourses fails to abide by the principle of reciprocity, which is a fundamental principle of international law and international relations. | -0.50 | 27 | -0.58 | 26 | -0.45 | 30 |
| 11 | The principle of equitable and reasonable utilization in current international water law has not progressed beyond the preliminary levels of simply outlining water quantity allocation and water use division. | -1.05 | 31 | 0.10 | 19 | -0.32 | 27 |
| 12 | For projects that may cause significant transboundary harm, countries that share transboundary rivers shall fulfill international obligations to avoid, contain and mitigate such harm. | 1.82 | 2 | 0.26 | 18 | 0.96 | 7 |
| 13 | Although "benefit sharing" is frequently mentioned in current international law, this concept lacks a uniform cognition, has not been fully established, and cannot be practically implemented. | -0.21 | 22 | 0.67 | 11 | 1.09 | 5 |
| 14 | Water benefit sharing theory requires that water stakeholders to be considered as a whole, with the purpose of promoting the maximization of benefits from the use of transboundary water resources, and emphasizing that all stakeholders can obtain the best benefits. | 1.25 | 5 | -0.96 | 29 | 1.19 | 4 |
| 15 | The core idea of water benefit sharing theory is not to allocate the actual water quantity, but to share the benefits obtained through the development and utilization of water resources. | 0.14 | 16 | -0.22 | 23 | -0.80 | 31 |
| 16 | The ideal water benefit sharing system is a positive-sum game, rather than a zero-sum game where only the amount of water available is allocated. | 0.44 | 11 | -0.38 | 25 | -0.10 | 22 |
| 17 | The benefits of water benefit sharing include environmental benefits to the water resources, such as improvement in water quality, ecological diversity, and environmental sustainability. | 0.44 | 12 | 0.76 | 8 | 0.58 | 11 |
| 18 | The benefits of water benefit sharing include direct economic benefits obtained from the uses of water resources, such as hydropower development, agricultural irrigation, and navigation convenience. | 0.43 | 13 | -1.46 | 36 | 0.47 | 12 |
| 19 | The benefits of water benefit sharing include political benefits related to water management, such as the reduction of political costs due to the resolution of international conflicts or the enhancement of international collaboration. | 0.73 | 8 | 1.88 | 2 | 0.45 | 13 |
| 20 | The benefits of water benefit sharing include indirect benefits beyond the water sector, such as promoting infrastructure construction, growth in trade, and so on. | -1.79 | 36 | 2.12 | 1 | -1.39 | 33 |
| 21 | Among the various benefits gained from water benefit sharing, environmental benefits to the water resources should form the basis of other types of benefits. Environmental benefits will actively promote other types of benefits, while the reverse is not necessarily true. | -0.44 | 26 | 1.39 | 3 | 0.23 | 18 |
| 22 | The benefit distribution of the water benefit sharing model could take various forms of equal distribution, proportional distribution according to the required project cost, or equitable distribution of different types of benefits. | 0.13 | 17 | 0.57 | 12 | 0.63 | 9 |
| 23 | Although the principle of benefit sharing does not violate any existing principle of current international water law, it has not become an explicit obligation of any party. | -1.16 | 33 | 1.09 | 5 | -0.00 | 21 |

(*Continued*)

**Table 2.** (Continued)

| NO. Q | Q Statements | F2 (Supporters) | | F5 (Idealists) | | F6 (Pragmatists) | |
|---|---|---|---|---|---|---|---|
| | | Score | Order | Score | Order | Score | Order |
| 24 | In the context where the principle of good-faith cooperation in international water law applies, countries should at least seriously consider benefit-sharing arrangements proposed by other countries. | 0.63 | 9 | 0.68 | 10 | 1.47 | 2 |
| 25 | The idea of water benefit sharing includes the calculation of various aspects and the consideration of relevant benefits. These include for instance, water management in industry, agriculture, domestic uses and ecological protection, which could be understood by the development of a thorough and operable index system of water benefit distribution. | 0.91 | 6 | 0.05 | 21 | 0.32 | 15 |
| 26 | Due to the lack of any uniform standard or understanding, the concept of water benefit sharing is often ineffective in practice, achieving little more than some hollow "win-win" rhetoric. | -0.59 | 28 | 0.30 | 17 | -1.71 | 34 |
| 27 | In the development, use and protection of transboundary river resources, countries that utilize water resources and receive ecological benefits shall correspondingly compensate countries that protect these resources and the ecological environment. | -0.98 | 29 | 1.19 | 4 | -0.43 | 29 |
| 28 | In the process of sharing international rivers, ecological benefit compensation should be established as a principle to guide the development, use and protection of transboundary rivers. | -0.00 | 20 | 0.32 | 16 | 0.30 | 16 |
| 29 | Countries may contribute to the ecological resources of transboundary rivers by taking active measures to protect resources and the ecological environment, such as forestation and the establishment of natural conservation areas. | 2.21 | 1 | 0.06 | 20 | 1.24 | 3 |
| 30 | Countries may contribute to the ecological resources of transboundary rivers by spontaneously restricting or refraining from certain activities, such as withdrawing from proposed dam construction, halting construction or expansion of industrial or mining operations, or the reduction of forest logging activities. | 1.73 | 3 | 0.34 | 15 | 0.60 | 10 |
| 31 | If the beneficiary countries of ecological resources in transboundary rivers do not provide reasonable compensation to those countries making contributions, it will probably dampen the enthusiasm in protecting transboundary water resources and ecological environment and negatively impact upon the improvement of relations between riparian states. | -0.21 | 23 | 0.79 | 7 | -0.41 | 28 |
| 32 | Ecological benefit compensation will address the shortcomings of the existent "polluter pays" principle and stimulate the protection of transboundary rivers and ecological environment. | 0.35 | 14 | -0.66 | 27 | 0.37 | 14 |
| 33 | Ecological benefit compensation of transboundary rivers ecology may take various forms including financial compensation, material compensation, and policy support. | 0.76 | 7 | 0.40 | 14 | 1.79 | 1 |
| 34 | Under the ecological benefit compensation principle, countries that contribute to the protection of ecological resources and the environment will be motivated by the appropriate compensation they receive, in furthering their protection activities. | 1.31 | 4 | 0.43 | 13 | 0.97 | 6 |
| 35 | Under the ecological benefit compensation principles, beneficiary countries that have paid compensation will further appreciate the preciousness of such resources and the ecological environment that they had to pay to preserve. | 0.63 | 10 | 0.81 | 6 | 0.18 | 20 |
| 36 | The ecological benefit compensation principle will facilitate the elevation from individual interest among riparian states to maximizing the collective benefits across the region. | -0.00 | 21 | -0.15 | 22 | -0.96 | 32 |
| 37 | Any dispute in relation to the ecological benefit compensation principle could be resolved through bilateral or multilateral discussion and negotiation, failing which there may be a judicial recourse to the International Court of Justice. | -1.05 | 30 | 0.76 | 9 | 0.82 | 8 |

party in the context of international water law (S23). They firmly support the 'no significant harm' rule of international water law in transboundary water projects (S12), and incidentally disagree with most of the criticism against the current law and the Convention on the Law of the Non-navigational Uses of International Watercourses (UNWC), such as that the no-harm rule favours downstream countries (S8), the failure of the UNWC in upholding fairness by favouring downstream countries and imposing excessive obligations on upstream countries (S9), and the lack of reciprocity in the UNWC (S10). This stream would most readily dismiss any attempt to emphasise the disparity between the rights and obligations of riparian states due to the regional particularity of transboundary rivers (S1).

In terms of the details of benefit sharing, this stream of views prefers a positive-sum game, rather than a zero-sum game where only water quantity is allocated (S16). These participants agree with most types of benefits, including environmental benefits to the water itself (S17),

Table 3. Stances on notable issues by perspective grouping.

| Key elements of benefit sharing | Statements | Supporters (Southeast Asia) | Idealists (South Asia) | Pragmatists (China) |
|---|---|---|---|---|
| The prospect of establishing benefit sharing principle | Support for 'no significant harm' principle | Strongly agree | Neutral | Agree |
| | Benefit sharing cannot be practically implemented | Neutral | Agree | Strongly agree |
| | Water benefit sharing should maximize benefits for all stakeholders | Strongly agree | Disagree | Strongly agree |
| | Benefit sharing only leads to hollow 'win-win' rhetoric | Disagree | Neutral | Strongly disagree |
| | Benefit sharing is not part of explicit obligations of any party | Strongly disagree | Strongly agree | Neutral |
| The definition of benefits | Water benefit sharing should include environmental benefits | Agree | Agree | Agree |
| | Water benefit sharing should include direct economic benefits | Agree | Strongly disagree | Agree |
| | Water benefit sharing should include political benefits | Agree | Strongly agree | Agree |
| | Water benefit sharing should include indirect benefits | Strongly disagree | Strongly agree | Strongly disagree |
| | Environmental benefits should form the basis of all other types of benefits | Disagree | Strongly agree | Neutral |
| Ecological compensation | Countries that use resources should compensate countries that make contributions to the environment | Disagree | Strongly agree | Disagree |
| | If the beneficiary countries of ecological resources in transboundary rivers do not provide reasonable compensation to those countries making contributions, it will probably dampen the enthusiasm in protecting transboundary water resources and ecological environment and negatively impact upon the improvement of relations between riparian states. | Neutral | Agree | Disagree |

direct economic benefits such as hydropower or navigation convenience (S18), and political benefits such as reducing conflicts and enhancing collaboration (S19). However, they strongly reject the inclusion of indirect benefits beyond the water sector, such as growth in trade or the development of infrastructure (S20). This group are also the least supportive for prioritizing environmental benefits above the other types (S21). They strongly believe that contribution to the transboundary river resources should include both active measures of protecting the environment, such as forestation (S29), and refraining measures, such as withdrawing construction or reducing logging activities (S30). This stream believes that any ecological compensation mechanism will incentivize further protection activities (S34). This group are the most confident in establishing an operable index system of water benefit distribution (S25). However, they are also the least supportive of any suggestion that disputes could be resolved by the International Court of Justice if these are not settled by multilateral discussion and negotiation (S37).

Overall, supporters largely agree with the current theories and values of international water law and the main conceptions of benefit sharing and ecological benefit compensation. People with such views are more inclined towards mutual gains for all water stakeholders that take into account multiple types of benefits, but without necessarily emphasising environmental benefits as of paramount importance; though they are against the inclusion of non-water sector indirect benefits into the calculation. The belief here is that different benefits can be managed in an index system, with ecological benefits compensation to take into account various contributions by riparian states and to motivate further protection activities.

## 4.2 Idealists

The second stream of views appear far less approving of some of the current theories than those supporters in the first stream. In sharp contrast to the other groups, participants from this stream disagree with the basic notion that benefit sharing requires the consideration of all

**Table 4. Notable views on Q statements by groups.**

| | Supporters | Score |
|---|---|---|
| Most agreed | Countries may contribute to the ecological resources of transboundary rivers by taking active measures to protect resources and the ecological environment, such as forestation and the establishment of natural conservation areas. (S29) | **2.21** |
| | For projects that may cause significant transboundary harm, countries that share transboundary rivers shall fulfil international obligations to avoid, contain and mitigate such harm. (S12) | **1.82** |
| | Countries may contribute to the ecological resources of transboundary rivers by spontaneously restricting or refraining from certain activities, such as withdrawing from proposed dam construction, halting construction or expansion of industrial or mining operations, or the reduction of forest logging activities. (S30) | **1.73** |
| | Under the ecological benefit compensation principle, countries that contribute to the protection of ecological resources and the environment will be motivated by the appropriate compensation they receive, in furthering their protection activities. (S34) | **1.31** |
| | Water benefit sharing theory requires that water stakeholders to be considered as a whole, with the purpose of promoting the maximization of benefits from the use of transboundary water resources, and emphasizing that all stakeholders can obtain the best benefits. (S14) | **1.25** |
| Most disagreed | The right to access and use of international rivers should not be referred to as water right. Instead, it should be regarded as part of the sovereignty. (S3) | **-1.88** |
| | The benefits of water benefit sharing include indirect benefits beyond the water sector, such as promoting infrastructure construction, growth in trade, and so on. (S20) | **-1.79** |
| | Among the conflicts that arose from countries that share international rivers, the more cooperation mechanisms there are, the harder it is to coordinate interests. (S7) | **-1.69** |
| | Due to the regional particularity of transboundary rivers, there is often considerable disparity between the rights and obligations for each individual riparian state. (S1) | **-1.28** |
| | Although the principle of benefit sharing does not violate any existing principle of current international water law, it has not become an explicit obligation of any party. (S23) | **-1.16** |
| | **Idealists** | **Score** |
| Most agreed | The benefits of water benefit sharing include indirect benefits beyond the water sector, such as promoting infrastructure construction, growth in trade, and so on. (S20) | **2.12** |
| | The benefits of water benefit sharing include political benefits related to water management, such as the reduction of political costs due to the resolution of international conflicts or the enhancement of international collaboration. (S19) | **1.88** |
| | Among the various benefits gained from water benefit sharing, environmental benefits to the water resources should form the basis of other types of benefits. Environmental benefits will actively promote other types of benefits, while the reverse is not necessarily true. (S21) | **1.39** |
| | In the development, use and protection of transboundary river resources, countries that utilize water resources and receive ecological benefits shall correspondingly compensate countries that protect these resources and the ecological environment. (S27) | **1.19** |
| | Although the principle of benefit sharing does not violate any existing principle of current international water law, it has not become an explicit obligation of any party. (S23) | **1.09** |
| Most disagreed | The right to access and use of international rivers should not be referred to as water right. Instead, it should be regarded as part of the sovereignty. (S3) | **-2.30** |
| | The benefits of water benefit sharing include direct economic benefits obtained from the uses of water resources, such as hydropower development, agricultural irrigation, and navigation convenience. (S18) | **-1.46** |
| | Among the conflicts that arose from countries that share international rivers, the more riparian states there are, the greater the differences of interests. (S5) | **-1.40** |
| | The allocation of water right is closely related to a country's national strategy. (S2) | **-1.38** |
| | International water rights over shared transboundary rivers are specific sharing schemes negotiated among riparian states, rather than a certain kind of right or interest that is well-documented, regulated, and predictable. (S4) | **-1.31** |
| | **Pragmatists** | **Score** |

**Table 4.** (Continued)

| Most agreed | Ecological benefit compensation of transboundary rivers ecology may take various forms including financial compensation, material compensation, and policy support. (S33) | **1.79** |
|---|---|---|
| | In the context where the principle of good-faith cooperation in international water law applies, countries should at least seriously consider benefit-sharing arrangements proposed by other countries. (S24) | **1.47** |
| | Countries may contribute to the ecological resources of transboundary rivers by taking active measures to protect resources and the ecological environment, such as forestation and the establishment of natural conservation areas. (S29) | **1.24** |
| | Water benefit sharing theory requires that water stakeholders to be considered as a whole, with the purpose of promoting the maximization of benefits from the use of transboundary water resources, and emphasizing that all stakeholders can obtain the best benefits. (S14) | **1.19** |
| | Although "benefit sharing" is frequently mentioned in current international law, this concept lacks a uniform cognition, has not been fully established, and cannot be practically implemented. (S13) | **1.09** |
| Most disagreed | Among the conflicts that arose from countries that share international rivers, the more cooperation mechanisms there are, the harder it is to coordinate interests. (S7) | **-2.48** |
| | International water rights over shared transboundary rivers are specific sharing schemes negotiated among riparian states, rather than a certain kind of right or interest that is well-documented, regulated, and predictable. (S4) | **-2.26** |
| | The right to access and use of international rivers should not be referred to as water right. Instead, it should be regarded as part of the sovereignty. (S3) | **-1.80** |
| | Due to the lack of any uniform standard or understanding, the concept of water benefit sharing is often ineffective in practice, achieving little more than some hollow "win-win" rhetoric. (S26) | **-1.71** |
| | The benefits of water benefit sharing include indirect benefits beyond the water sector, such as promoting infrastructure construction, growth in trade, and so on. (S20) | **-1.39** |

water stakeholders as a whole to maximize benefit and to ensure that all stakeholders can obtain the best benefits (S14). Again, unlike the other groups, they do not disagree with the criticism that benefit sharing currently only achieves hollow 'win-win' rhetoric due to the lack of uniform standard or understanding (S26). They strongly disagree with the close association of allocation of water rights with a country's national strategy (S2). They also object to the suggestion that international water rights are dependent on specific sharing schemes negotiated by states rather than being a set of well-documented, regulated and predictable kind of right or interest (S4).

The most notable differences for this stream of views as compared with the others lie in the understanding of the details of benefit sharing. In sharp contrast to the other groups, people sharing this stream of views are firmly against the inclusion of direct economic benefits from use of water resources into water benefits, such as those derived from hydropower, irrigation and navigation (S18). At the same time, they strongly advocate the inclusion of indirect benefits such as the development of infrastructure and growth of trade, which the other groups clearly reject (S20). Combining with the fact that this is the only group which support the understanding that environmental benefits should form the basis of all other types of benefits (S21), the environmentally focused approach to the conception of water benefits becomes a prominent distinction for this stream of views.

The environmental focus of this stream is further evidenced in that this is the only group that support ecological compensation to be paid by countries who utilize water resources and receive ecological benefits to countries that protect these resources and the environment (S27). It is believed that not paying compensation would dampen the enthusiasm of countries that protect the environment (S31). Nevertheless, the intention here is not to change the "polluter

pays" approach (S32), but to make countries better appreciate the preciousness of the resources they have to pay to utilize (S35).

It may be observed that this stream of views seems highly idealistic in its focus on the environment above all others. Participants in this stream show little support for the practicality of contribution or measurement of contribution. Neither positive actions such as forestation (S29) nor refraining decisions such as withdrawing construction (S30) are viewed as contribution to ecological resources by this group, in contrast to others. Unlike supporters in the first stream above, idealists of this second stream do not agree with the establishment of any index system for water benefits distribution (S25). Even more notable is their belief that benefit sharing do not form the explicit obligations of any party under the current international law (S23).

Overall, idealists see environmental benefits as the basis of all types of water benefits. Utilizing ecological resources should entail compensation for those who contribute, to avoid dampening enthusiasm for protection as well as encouraging appreciation of the preciousness of the resources. There is the firm belief to exclude direct economic benefits from the calculation of water benefits, despite that, conceptually and practically, synergies can be developed to understand how best to harvest benefits from water, which serve industry, agriculture and households as well as ecological protection [53]. Idealists appear less concerned by the practical measures to contribute to ecological resources or the observation that benefit sharing is not an explicit obligation of any party. The belief seems to be that the fundamental importance of environmental benefits and the preciousness of ecological resources would be sufficient to persuade users of such resources to compensate those who made contribution.

## 4.3 Pragmatists

The third and final stream of views tend to be more pragmatic and often occupy the middle ground between supporters and the idealists. Participants in this stream most firmly reject the suggestion that international water rights are dependent on specific negotiation rather than being well-documented and predictable (S4). They clearly see the value of cooperation mechanisms in coordinating interests among conflicts over international rivers (S7). At the same time, they are the least resistant against any criticism of the current international law, such as that the principles of equitable uses and no significant harm benefit upstream and downstream countries differently (S8). On the fundamental perception of benefit sharing, this stream is largely undecided as to whether it is a part of the explicit obligations of any party under international law (S23), while practical implementation of the concept is seen with the greatest difficulty amongst all groups due to the lack of uniform cognition and establishment (S13). This has not stopped people holding views in this stream to continue to believe that benefit sharing should at least be seriously considered under the principle of good-faith cooperation (S24), or that benefit sharing could achieve more than hollow 'win-win' rhetoric (S26).

As to the practical details of benefit sharing, this stream is much closer in line with the supporters rather than the idealists. Environmental benefits, direct economic benefits and political benefits are all included in the calculation (S17, S18 & S19), but indirect benefits are excluded (S20). Unlike the idealists, environmental benefits are not afforded any special significance above other types of benefits (S21). This stream is much less pronounced with regard to the purposes of mechanisms of ecological compensation, whether to avoid dampening the enthusiasm for environmental protection (S31), or to promote the appreciation of the preciousness of the resources through having to pay for the usage (S35). They are the only group to seriously doubt whether ecological benefit compensation could facilitate the maximization of collective interests among all riparian states (S36). Nevertheless, this group has the clearest preference

for ecological benefit compensation to take various forms, including financial compensation, material compensation and policy support (S33).

Overall, the pragmatists believe in the value of international cooperation and benefit sharing in general. Yet they see great difficulty in many of the practical aspects for the recognition and implementation of such conception, especially the less established idea of ecological benefit compensation. This dual perception of soundness in principle and difficulty in practice seems to be the drive behind much of the preference for well-intentioned cooperation and consideration as well as the reticence on issues such as whether benefit sharing is an obligation of parties or the exact objectives behind ecological benefit compensation.

### 4.4 Impact of geographic locations

An important finding of this study after factor analysis is that the geographic association of the participants have significant impact on their views, even though they were not distinguished by such characteristics in the earlier part of the study. As illustrated by Table 1, the three streams of views represented by the three relevant factors of the Q-Method survey display strong geographic associations. Using the labels employed by the analysis above, all pragmatists are from China. 67% of the supporters are from Southeast Asia, while 60% of the idealists are from South Asia.

Such information of geographic association could make some of the interpretation easier to align with the existing hydrodiplomacy of the countries concerned. Take pragmatists in this study as an example, all of whom come from China as we now know. Their strong preferences for cooperation mechanisms would make sense given the efforts of China in promoting such cooperation in the region, such as the Lancang-Mekong Cooperation Mechanism (LMCM). Here, a 'win-win' rhetoric in relation to transboundary water sharing has been a favoured narrative of China [54], which also explains why the pragmatists would strongly disagree with any dismissal of the value of such cooperation. Indeed, China's approach in exploiting Mekong resources has long been criticized both for their environmental impacts on the local sustainability and for their lack of transparency in decision-making process [55]. Within the LMCM, the priority areas of cooperation cover a broad range of issues apart from water management, including "connectivity, production capacity cooperation, cross-border economic cooperation as well as agriculture and poverty-reduction cooperation". These are aligned with areas of cooperation among the ASEAN countries, which address how human needs can be satisfied by utilizing shared water resources. It is also noticeable that a strong emphasis on political stability in the region from China's side parallels with China's commitment to providing dedicated financial transactions within member countries for regional development [56].

### 4.5 Consensus and contention

Despite the analysis above of the three different main streams of views, there is notable commonality in the views of the majority of participants in this exercise, indicating an encouraging level of consensus on several general issues (Tables 4 & 5). Most notably, the notion that the right to access and use international rivers should be regarded as a part of sovereignty rather than water rights [57, 58] is universally and fiercely rejected by all, including experts from China. This is despite the debate as to whether China claim absolute or limited territorial sovereignty in transboundary waters as an 'upstream controller' [59]. Water rights have endowed the countries better form when seeking to secure profits from the shared water resources, hence a common ground when nation states aim to develop hydrodiplomacy. Therefore, in terms of the detailed understanding of benefit sharing, such agreement supports the idea that states act as lead representative in hydrodiplomacy when defining key benefits in water

sharing. Participants disagree with the assertion that more cooperation mechanism render it harder to coordinate interests among countries [58, 60], though to varying extents among the three streams of views. Furthermore, there is consistent level of support for principles such as 'no significant harm' under current international law and UNWC among the participants. Some of the criticism against UNWC, such as that it failed to uphold the principle of fairness by favouring downstream countries over upstream countries [58, 61] are rejected by most.

To some extent, consensus is also seen in participants' agreeing with ecological compensation mechanism when practising benefit sharing principle. Ecological compensation, though may appear in various forms, essentially serves to compensate economic loss when countries contribute to ecological protection activities. Although this is seen by idealists to potentially burden countries' economy and development agenda (S35, S27), there is flexibility for it to take various forms, including financial compensation, material compensation and policy support (S33). Moreover, both supporters and pragmatists agree with adopting ecological compensation mechanism as an approach to compromise to maximise profits in the sharing of desired benefits (S30).

However, there are significant differences on the crucial issues of what should be included as benefits to be shared in transboundary rivers and what mechanisms facilitate the fair implementation of the principle. Participants generally agree on some of the basic understanding, such as that water benefit sharing is more about sharing economic and other benefits of water and national development rather than dividing a fixed quantity of water itself, which represents a narrow focused of the sustainable development of shared rivers [62]. As explained above, both supporters and pragmatists reject indirect benefits beyond the water sector, such as infrastructure development or growth in trade. Yet this type represents the most keenly desired benefits by idealists, who in turn strongly reject direct economic benefits such as hydropower or irrigation, which are comfortably accepted by others. Although current scholarship often promotes environmental benefits as the basis of all other types of benefits [9, 13], this turns out to be one of the most controversial topics among all participants (Table 5). There is considerable disagreement as to whether refraining activity, such as withdrawing from construction project or reducing mining or logging operations, should be seen as contribution to the ecological resources of transboundary rivers [43].

In addition, disagreements also exist on how to share the desired benefits, including mechanisms that facilitate the fair distribution of benefits as well as institutions for conflict resolution. Despite the understanding that distribution of benefits could take various forms such as equal distribution, proportional distribution according to cost or equitable distribution of different types of benefits [63], it may be observed that the equitable use of water resources pose challenge to countries because of their different water needs shaped by social and economic development and the regional particularity of transboundary rivers [64]. Some disagreement exists as to whether the existing international norm facilitates countries to forming effective cooperative mechanisms. Participants fundamentally disagree on whether disputes arising out of benefit sharing and ecological benefits compensation could be resolved through the International Court of Justice, as suggested by some [43].

## 5. Discussions and implications of the findings

### 5.1 Regional perspectives in sharing international rivers

The most significant insight to be gained from the findings of this Q-Method survey is the starkly different perspectives with regard to the types of benefits to be shared over international rivers. Such novel understanding based on empirical evidence highlights concerns raised

**Table 5. Most and least controversial Q statements among all participants.**

| | All participants | Standard Deviation |
|---|---|---|
| **Most controversial** | Among the conflicts that arose from countries that share international rivers, the greater the differences of national conditions (e.g. socio-economics) there are, the more claims for interests there will be. (S6) | **1.816** |
| | Any dispute in relation to the ecological benefit compensation principle could be resolved through bilateral or multilateral discussion and negotiation, failing which there may be a judicial recourse to the International Court of Justice. (S37) | **1.811** |
| | Due to the regional particularity of transboundary rivers, there is often considerable disparity between the rights and obligations for each individual riparian state. (S1) | **1.732** |
| | The benefits of water benefit sharing include indirect benefits beyond the water sector, such as promoting infrastructure construction, growth in trade, and so on. (S20) | **1.662** |
| | Among the various benefits gained from water benefit sharing, environmental benefits to the water resources should form the basis of other types of benefits. Environmental benefits will actively promote other types of benefits, while the reverse is not necessarily true. (S21) | **1.632** |
| **Least controversial** | Ecological benefit compensation will address the shortcomings of the existent "polluter pays" principle and stimulate the protection of transboundary rivers and ecological environment. (S32) | **1.314** |
| | The core idea of water benefit sharing theory is not to allocate the actual water quantity, but to share the benefits obtained through the development and utilization of water resources. (S15) | **1.288** |
| | For projects that may cause significant transboundary harm, countries that share transboundary rivers shall fulfil international obligations to avoid, contain and mitigate such harm. (S12) | **1.083** |
| | The benefit distribution of the water benefit sharing model could take various forms of equal distribution, proportional distribution according to the required project cost, or equitable distribution of different types of benefits. (S22) | **1.067** |
| | The Convention on the Law of the Non-Navigational Uses of International Watercourses fails to strictly uphold the principle of fairness, in favouring the interests in water resources of the downstream countries and imposing excessive obligations on the upstream countries. (S9) | **1.052** |

in the implementing of benefit sharing principle. In Southeast Asia and South Asia, the features of geopolitics in the region strongly impact their understanding of the practice of benefit sharing while obscuring the underlying reasons that affect the reaching of agreements between states on benefit sharing. Power asymmetry is recognized to be significantly affecting the nature of international cooperation and tensions in the management of river basins in Asia [10], hence it is likely to constitute an important factor affecting countries' adopting preferred mechanism when practising the sharing of benefits from shared water resources. Concerns are raised how political power affects the reaching of just outcomes for each country to secure their rightful interests in the shared water resources, although both pragmatists and idealists agree that water rights can be negotiated through inter-state bilateral or multilateral discussion (S4). Similarly, the low economic development level in Asia and the underdevelopment of key democratic institutions as well as limited knowledge in water management have posed challenges to the region in setting up efficient institutional arrangements to promote sustainable development [65]. Those who are equipped with material power is more likely to be well-positioned when promoting ecological compensation mechanism that features the provision of financial or material compensation.

Indeed, political and economic asymmetry have obscured the understanding of Asian states' negotiating for governance arrangements over shared rivers, especially on powerful actors and their interactions with neighbouring states. For its sheer size and economic and political power, China is an influential actor impacting on the uses and governance of the shared water resources. With China's engaging with the Mekong countries mainly through economic profits rather than environmental protection in the river basin, the 'empirical neglect' of environmental benefits had been a major criticism against China's policy in the Mekong River [9]. This Q-Method survey results would indicate that such position may well be reciprocated downstream in Southeast Asian countries. Although they disagree on a number of other issues, both supporters, predominantly from Southeast Asia, and pragmatists, exclusively from China, agree on including environmental benefits, direct economic benefits and political benefits into the calculation of water benefits, but not infrastructure or trade related indirect benefits. Indeed, it may be argued that such largely compatible views among Chinese experts and their Southeast Asian neighbours on the practically significant issues of what to share, what not to share, and whether environmental benefits should be prioritized, potentially contribute to the advancement and progression in the cooperation of these countries, most clearly illustrated by the LMCM [56].

Such a level of compatibility may be far more difficult to achieve when working with the group of idealists identified in the present study, most of them from South Asia. Idealists focus on environmental benefits, as advocated by the leading theories. However, they also reject direct economic benefits such as hydropower generation or navigation convenience in the sharing of international river benefits. It may be inferred that idealists want to see countries voluntarily give environmental concerns the highest priority without being obliged, to pay for usage of such precious resource without having its contributions recognized, and to exclude direct economic benefits from their understanding and calculation. Such position has also been found in domestic water management in India, where the impact of environmental movement, which sees the involvement of a prominent idealist perspective, is significant in the policy process [27]. Some opine that environmental activism challenges national government's role to promote the management of water resources in a centralized fashion [27]. It seems highly doubtful whether any cooperation or sharing of international rivers so far could satisfy such high expectations of the idealists. Given the small sample base of experts from South Asia, this present study by no means indicates that this is a dominant stream of views in the region. It is nevertheless a notable and distinct perspective that deserve attention and scrutiny by future studies.

## 5.2 Framework of benefit sharing

Existing scholarship has placed more emphasis on the construction of benefit sharing than the importance of agreed benefit sharing principle in the development of hydrodiplomacy. As discussed above, the four types of benefits in sharing international rivers are widely accepted by international scholarship, and this paper has no intention to challenge such construction. Indeed, the latest literature often seeks to develop these four types further, for example by including 'social-cultural cooperation' into the category of indirect benefits 'beyond the river' [33]. Yet acceptance of such theoretical development by many expert participants in this study seems to lag behind, given that many would even disagree with including indirect benefits into water benefit sharing in the first place. At the same time, while few in this realm would explicitly question that environmental benefits should underpin all other benefits in the context of international rivers, it seems legitimate to investigate whether that belief is widely shared or firmly held by experts and officials in practice. As illustrated in this study, when placed

alongside other competing values and priorities, the foundational role of environmental benefits is readily downgraded by many.

Such subjective and regionally-associated perspectives of experts uncovered in this present study pave the way for further theorization of the concept of benefit sharing that takes into account the more practical and more localized aspects, in the complex context of the great varieties of international river sharing in the world. Without changing the typological framework first introduced by Sadoff and Grey [13], it is nevertheless sensible to take into consideration of new variables such as regional differences and economical and developmental status. Countries in different parts of the world may have very different social, economic, political and cultural understanding and expectation of using water resources [66]. In order for benefit sharing to thrive as a concept and a practical approach of international water governance, its theoretical basis should encapsulate the level of awareness and flexibility that would accommodate considerably different perspectives, such as those of regional experts identified in this study.

The accommodation of such perspectives will enhance the role and significance of the theory of benefit sharing, in explaining and analysing the success or difficulty of reaching agreements on international cooperation over water resources. With the increasing recognition of the importance of benefit sharing on the international stage, a more comprehensive, adaptable and multi-faceted theory of benefit sharing could offer invaluable insights that greatly enrich the current focus on considerations such as power or geopolitics.

## 5.3 Benefit Sharing in Hydrodiplomacy Process

The findings of this present study have shown that, compatible with outcomes of hydro- diplomatic practices, when countries have developed similar views on benefit sharing, they are likely to achieve positive cooperation on the sharing of international rivers. This is the case of the Mekong River basin, where views between the Chinese and the Mekong groups are closely aligned. In contrast, disagreement on benefit sharing principle among nation states, especially those in the Global South, constitutes an important factor to their failing to develop institutional mechanisms for water sharing. The starkly different views between Chinese and Indian experts on benefit sharing help to explain why hydrodiplomacy between China and India is fraught with difficulties. It is argued that countries' agreement on the conceptual framework of benefit sharing principle is an essential element in successful hydrodiplomacy.

While open discussion of the conceptual framework of benefit sharing is evidently helpful, often it is not placed on the top of political agenda in hydrodiplomacy, possibly because of countries' lacking appreciation of its importance. Nevertheless, for countries in the Global South, benefit sharing underlies their approaches in water management, which orients toward garnering material interests while achieving sustainable goals for the river basins. Neglecting or shying away from discussion of benefit sharing is obviously detrimental to any effort to reach consensus. This is a particularly pertinent concern because current international law has not placed clear obligations of incorporating benefit sharing on countries that intend to cooperate. Therefore, it is important for countries to address the basic issues of benefit sharing early in the process of hydrodiplomacy, such as the types of benefits to share and the acceptance of sharing mechanisms including ecological compensation.

## 6. Conclusion

Although benefit sharing is the buzz word of the current discourse on transboundary waters globally, there are notable difficulties in the conception and implementation of the approach. Based on Q-Method empirical findings, this article identifies and expounds the notable

differences in the subjective perspectives and understanding of benefit sharing among experts from China, South and Southeast Asia. Despite their differences on issues such as practical implementation of benefit sharing or whether it forms a part of parties' explicit obligations, supporters from Southeast Asia and pragmatists from China agree on several key issues such as the type of benefits to be included in sharing (environmental, direct economic and political benefits) and those to be excluded (indirect benefits), and do not see environmental benefits as forming the basis of all other types of benefits. Such compatibility of views potentially makes collaboration between the countries more feasible. On the other hand, idealists from South Asia put environmental benefits above all other benefits and strongly object to direct economic benefits. Such drastically different approaches to the understanding of benefits in sharing international rivers could be an underlying factor in the difficulty of reaching agreements between China and India for instance.

Furthermore, such understanding contributes to the conception of benefit sharing in two ways. Firstly, the current theory of benefit sharing should develop to encapsulate the variety of perspectives from different parts of the world. The social, economic and political conditions in Asia, for example, differ greatly from those in Western Europe or North America, where much of the theoretical framework of benefit sharing originates from. Such differences and their implications on the conception of benefit sharing in developing countries such as different regions of Asia should be afforded more attention in future advancement of the concept, so that it would facilitate a wider range of international cooperation and agreements ingrained with the ethos of benefit sharing. Secondly, such a broader approach would in turn enhance the value of the institution of benefit sharing in understanding environmental governance and hydrodiplomacy, to the extent that existing success stories as well as obstacles and difficulties of sharing international rivers could be better analysed and interpreted incorporating the approach of benefit sharing. It is foreseeable that, with the conjoined efforts of theorists and practitioners in the area, benefit sharing could realize its full potential as an integral approach to solving disputes and maximizing benefits over the limited and precious resources of our shared world.

## Supporting information

**S1 File. Benefit sharing in Asia.**
(PDF)

## Author Contributions

**Conceptualization:** Lei Xie.

**Data curation:** Qi Yu.

**Formal analysis:** Lei Xie, Lu Xu, Qi Yu.

**Funding acquisition:** Lei Xie.

**Investigation:** Lei Xie.

**Methodology:** Lei Xie, Lu Xu.

**Software:** Qi Yu.

**Writing – original draft:** Lei Xie, Lu Xu.

**Writing – review & editing:** Lei Xie, Lu Xu.

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
