## [Decision Letter · Decision Letter 0]

5 Dec 2022

PONE-D-22-27342Benefit sharing in international rivers: A Q-Methodology study of regional understanding and perception in AsiaPLOS ONE

Dear Dr. Xu,

Thank you for submitting your manuscript to PLOS ONE. After careful consideration, we feel that it has merit but does not fully meet PLOS ONE’s publication criteria as it currently stands. Therefore, we invite you to submit a revised version of the manuscript that addresses the points raised during the review process.

We look forward to receiving your revised manuscript.

Kind regards,

László Vasa, PhD

Academic Editor

PLOS ONE

Journal Requirements:

"Lei Xie received funding from Shandong University (www.sdu.edu.cn, Funding Reference No.61060082035302). The funder had no role in study design, data collection and analysis, decision to publish, or preparation of the manuscript."

We note that one or more of the authors is affiliated with the funding organization, indicating the funder may have had some role in the design, data collection, analysis or preparation of your manuscript for publication; in other words, the funder played an indirect role through the participation of the co-authors. If the funding organization did not play a role in the study design, data collection and analysis, decision to publish, or preparation of the manuscript and only provided financial support in the form of authors' salaries and/or research materials, please do the following:

a. Review your statements relating to the author contributions, and ensure you have specifically and accurately indicated the role(s) that these authors had in your study. These amendments should be made in the online form.

b. Confirm in your cover letter that you agree with the following statement, and we will change the online submission form on your behalf: 

“The funder provided support in the form of salaries for authors [insert relevant initials], but did not have any additional role in the study design, data collection and analysis, decision to publish, or preparation of the manuscript. The specific roles of these authors are articulated in the ‘author contributions’ section.”"

Reviewers' comments:

Reviewer's Responses to Questions

**Comments to the Author**

1. Is the manuscript technically sound, and do the data support the conclusions?

Reviewer #1: Yes

Reviewer #2: Partly

2. Has the statistical analysis been performed appropriately and rigorously? 

Reviewer #1: Yes

Reviewer #2: Yes

3. Have the authors made all data underlying the findings in their manuscript fully available?

Reviewer #1: Yes

Reviewer #2: Yes

4. Is the manuscript presented in an intelligible fashion and written in standard English?

Reviewer #1: Yes

Reviewer #2: Yes

5. Review Comments to the Author

Reviewer #1: The paper is addressing a complicated and widely disputed problem of cooperation on transboundary water resources: benefit sharing and ecological benefit compensation. The chosen Q methodology makes possible the objective and impartial description of perceptions and views of researchers, civil society actors and decision makers without the authors having to take a firm position on issues on which there is a wide-ranging and far from conclusive debate is going on within the academic community. The paper makes a useful contribution to this debate by highlighting the necessity of further progress on reconciling potentially competing interests over shared water resources. The significant differences among the three "strains of views" identified by the authors reveal the fact that international water law (and here the authors could have mentioned the 1992 UNECE Water Convention, to which any state can accede now, i.e. which is an authoritative document on international water law) takes too narrow an approach. The authors correctly point out that at present international water law does not offer any solid guidance on benefit sharing and environmnetal benefit compensation. The resulting, often contraditory viewes of experts identified by the Q methodology reveals that loopholes in international water law make it more difficult to negotiate bilateral or multilateral agreements on the join, cooperative management of shared transboundary water resources. The paper, by highlighting these differences, usefully contributes to the ongoing discussion on the need to further develop international water law so it would provide more guidance than at present on benefit-sharing. The paper presents in a balance manner an impressive number of publications on benefit sharing and ecological benefit compensation without passing a judgement on concrete, disputed issues thanks to the use of Q methodology. The use of Q methodology is particulary useful for facilitating further progess in water diplomacy, as experts and decision makers that prepare negotiating strategies need to be aware of the percetions and understandings of the negotiating partners, beyond a thorough knowledge of specific problems. The main strength of the paper is its innovative use of Q methodology. At the same time, the methodology also limits the depth of analysis. The paper provides only a snapshot of the present positions of a limited number of a (not necessarily fully representative group of) experts from a limited number a countries. While it mentions significant developments in the position of China on international water cooperation, it does not do the same for the water cooperation policies of neighbouring countries. To sum up: analyzing the views and perceptions of experts on benefit sharing and ecological benefit compensation from three geographical regions (China, South-East Asia and South Asia) using the Q methodology successfully draws attention to the need for the further development of international water law and for a more thorough theoretical and professional grounding of hydro-diplomacy. At the same time, the differences of opinion revealed by the Q methodology clearly indicate that experts in benefit-sharing still have considerable work to do. The growing water scarcity caused by climate change makes it more urgent than ever to intensify research and scientific exchange in this field.

Reviewer #2: The abstract should be more compact; the results should be indicated as well. Among the key words, the geographical locations should be mentioned as well ("e.g. "China").

In the 2. Chapter ("Conceptualization...") it is not clear, why a specific field ("2.1 Contextualizing benefit sharing in Asia") is discussed before a general one ("2.2 Definitions of benefits" - BTW, this title is too general, benefits of what?).

While I fully agree with the authors' choice to make a conceptualization, the "real" literature review is missing. I recommend to include an overview of the literature of the topic, includong non-Asian focused papers as well.

Recommendations and implications are not defined within the conclusion chapter. Limitations should be described as well.

6. PLOS authors have the option to publish the peer review history of their article (what does this mean?). If published, this will include your full peer review and any attached files.

Reviewer #1: **Yes: **Márton Krasznai

Reviewer #2: No

---

## [Author Response · Author response to Decision Letter 0]

7 Dec 2022

Response uploaded as separate file.

---

## [Editor Report · Decision Letter 1]

5 Jan 2023

Benefit sharing in international rivers: A Q-Methodology study of regional understanding and perception in Asia

PONE-D-22-27342R1

Dear Dr. Xu,

We’re pleased to inform you that your manuscript has been judged scientifically suitable for publication and will be formally accepted for publication once it meets all outstanding technical requirements.

Kind regards,

László Vasa, PhD

Academic Editor

PLOS ONE
---

## [Editor Report · Acceptance letter]

8 Jan 2023

PONE-D-22-27342R1 

Benefit sharing in international rivers:
A Q-methodology study of regional understanding and perception in Asia 

Dear Dr. Xu:

I'm pleased to inform you that your manuscript has been deemed suitable for publication in PLOS ONE. Congratulations! Your manuscript is now with our production department. 

Kind regards, 

on behalf of

Prof. Dr. László Vasa 

Academic Editor

PLOS ONE